# Kinetics of the Organic Compounds and Ammonium Nitrogen Electrochemical Oxidation in Landfill Leachates at Boron-Doped Diamond Anodes

**DOI:** 10.3390/ma14174971

**Published:** 2021-08-31

**Authors:** Barbara Krystyna Wilk, Małgorzata Szopińska, Aneta Luczkiewicz, Michał Sobaszek, Ewa Siedlecka, Sylwia Fudala-Ksiazek

**Affiliations:** 1Department of Water and Wastewater Technology, Faculty of Civil and Environmental Engineering, Gdansk University of Technology, 11/12 Narutowicza St., 80-233 Gdansk, Poland; malszopi@pg.edu.pl (M.S.); aneta.luczkiewicz@pg.edu.pl (A.L.); 2Department of Metrology and Optoelectronics, Faculty of Electronics, Telecommunication and Informatics, Gdansk University of Technology, 11/12 Narutowicza St., 80-233 Gdansk, Poland; michal.sobaszek@pg.edu.pl; 3Faculty of General and Inorganic Chemistry, University of Gdansk, Wita Stwosza 63 St., 80-306 Gdansk, Poland; ewa.siedlecka@ug.edu.pl; 4Department of Sanitary Engineering, Faculty of Civil and Environmental Engineering, Gdansk University of Technology, 11/12 Narutowicza St., 80-233 Gdansk, Poland; sksiazek@pg.edu.pl

**Keywords:** boron-doped diamond electrode (BDD), advanced oxidation processes, electrooxidation (EO) kinetics, optimization of energy consumption

## Abstract

Electrochemical oxidation (EO) of organic compounds and ammonium in the complex matrix of landfill leachates (LLs) was investigated using three different boron-doped diamond electrodes produced on silicon substrate (BDD/Si)(levels of boron doping [B]/[C] = 500, 10,000, and 15,000 ppm—0.5 k; 10 k, and 15 k, respectively) during 8-h tests. The LLs were collected from an old landfill in the Pomerania region (Northern Poland) and were characterized by a high concentration of N-NH_4_^+^ (2069 ± 103 mg·L^−1^), chemical oxygen demand (COD) (3608 ± 123 mg·L^−1^), high salinity (2690 ± 70 mg Cl^−^·L^−1^, 1353 ± 70 mg SO_4_^2−^·L^−1^), and poor biodegradability. The experiments revealed that electrochemical oxidation of LLs using BDD 0.5 k and current density (j) = 100 mA·cm^−2^ was the most effective amongst those tested (C_8h_/C_0_: COD = 0.09 ± 0.14 mg·L^−1^, N-NH_4_^+^ = 0.39 ± 0.05 mg·L^−1^). COD removal fits the model of pseudo-first-order reactions and N-NH_4_^+^ removal in most cases follows second-order kinetics. The double increase in biodegradability index—to 0.22 ± 0.05 (BDD 0.5 k, j = 50 mA·cm^−2^) shows the potential application of EO prior biological treatment. Despite EO still being an energy consuming process, optimum conditions (COD removal > 70%) might be achieved after 4 h of treatment with an energy consumption of 200 kW·m^−3^ (BDD 0.5 k, j = 100 mA·cm^−2^).

## 1. Introduction

Despite a new solid waste strategy and enhanced recycling and utilization, landfilling still represents an important path for municipal solid waste management. Landfilling leads to the formation of landfill leachates (LLs), which are potentially toxic and complex liquids with high refractory contaminant levels that percolate through landfill [1,2,3]. LLs are most often classified by age. LLs from new cells (1–2 years old) usually have higher concentrations of biodegradable contaminants (ratio of five-day Biochemical Oxygen Demand to Chemical Oxygen Demand, BOD_5_/COD) ratio is >0.5) due to the high concentration of volatile organics) [4,5]. LLs generated by older cells (>10 years old) are usually less prone to biological degradation with BOD_5_/COD < 0.3 (due to the high concentration of refractory organic matter) [6]. However, the appropriate BOD_5_/COD ratio itself does not ensure good biochemical degradation of contaminants [7]. Therefore, each LL treatment problem requires a comprehensive, individual approach. The problem of efficient and economically viable treatment of LLs is now one of the crucial, still-unresolved environmental issues globally.

According to the relevant legal requirements [8,9,10], LLs should be collected and subsequently treated to ensure that they are safe for the environment. However, there are no unique and flexible treatment methods available as of now [11]. A number of technologies and processes for LL treatment have been assessed in recent decades, including biological processes, adsorption, coagulation/flocculation, the membrane process, and advanced oxidation processes (AOPs) [12]. Biological processes (conventional activated sludge, sequencing batch reactors, membrane bioreactors, aerated lagoons, and upflow anaerobic sludge blanket) are most frequently used for pre-treatment of LLs generated by new landfill cells. The aim of the aforementioned processes is the simultaneous removal of nitrogen and easily biodegradable organic matter from the LLs. However, in LLs, biological treatment is usually hampered by certain toxic substances and inadequate C-to-N ratio. The probable presence of non-biodegradable emerging pollutants is also of concern. The outcome is thus unsatisfactory and requires the inclusion of an additional treatment step to meet the discharging standards [13,14,15,16]. Recently, AOPs are considered to be among the most efficient methods in disinfecting and purifying ground water and natural water, as well as being used in wastewater treatment [17,18]. The processes are becoming increasingly popular, particularly in industrial settings, as they are generally accepted to be environmentally friendly and very much in line with the Clean Production strategy (minimizing the volume of contaminants released into the environment) [19,20,21]. Among the most efficient and fastest developing AOPs are the electrochemical ones [22,23], which include: anodic oxidation, anodic oxidation with electrogenerated H_2_O_2_, electro-Fenton, photoelectro-Fenton, and solar photoelectro-Fenton [24].

Among the EO process, reagent-free anodic oxidation is increasingly popular. The main objective of these processes is to generate free hydroxyl radicals (^^•^^OH), which are non-selective, powerful oxidizing agents capable of oxidizing a wide variety of contaminants at nearly diffusion-limited rates. Anodic oxidation is conducted with the use of high-oxygen-overvoltage anodes (so-called “non-active” anodes), which include, among others, SnO_2_, PbO_2_, Ti_4_O_7_, and BDD. BDD has many unique vital characteristics: a broad range of electrochemical potentials in aqueous solutions (from −1.25 to 2.3 V, as compared to a standard hydrogen electrode [SHE]). The use of EO to treat wastewater by means of BDD anodes was reported in earlier studies [25,26,27,28,29]. In accordance with Comninellis’ model, the degradation of organic compounds (R) in LLs by means of BDD anodes is mainly mediated by hydroxyl radicals through direct and/or indirect oxidation. In the case of direct oxidation, the contaminant first undergoes physical adsorption on the anode surface and is then oxidised via the intermediation of very reactive short-lived hydroxyl radicals (this occurs below the oxygen evolution reaction potential on the anodes) [30,31]. The indirect oxidation process is the consequence of the hydroxyl radicals (produced in the water discharge reaction) being physiosorbed in the vicinity of the anode surface [32], as shown in Equations (1) and (2).
H_2_O + BDD → BDD (^•^OH ) + H^+^ + 1e^−^(1)
BDD (^•^OH ) + R→ BDD + CO_2_ + H_2_O + H^+^ + 1e^−^(2)

Indirect anodic oxidation may also be mediated by electrogenerated oxidizing agents in the sample, such as hypochlorite, chlorine, ozone, or hydrogen peroxide [30]. As has been pointed out in the literature, the active chlorine removes N-NH_4_^+^ first [19,33,34]. Next, if there is surplus content of active chlorine in the LLs, intensive oxidation of organic compounds takes place. Most studies indicate [21,26,35] that the degradation of ammonium in LLs is a second-order reaction (first-order with respect to the concentration of ammonium, and first-order with respect to the concentration of active chlorine), while COD degradation fits a pseudo-first model, or pseudo-zeroth-order kinetics models. The kinetic model of COD degradation is determined by the following factors: contaminant concentration level in the analyzed matrix, the applied current density (j), the electrode substrate, pH and temperature [26,36,37]. The degradation of contaminants in the LLs matrix is a very complex process that has not yet been fully understood. On the other hand, the possible high electrolytic conductivity of LLs (e.g., high concentrations of chloride and sulfate) should benefit the efficiency of the EO process.

EO has also a great potential for medical sector application, especially in the case of hospital wastewater treatment rich in non-biodegradable pharmaceuticals (e.g., antibiotics, antihistaminics, anti-inflammatories, antidepressants, antihypertensives, hormones, and antiulcer agents) [38]. Moreover, transformation of urine into nutrients using electrolytic oxidation was also evaluated [33]. Additional positive side effect of EO wastewater treatment is disinfection, which is described by Dbira and co-authors [34] as well as Kraft [35].

The aim of the study was (I) to determine the efficiency of BDD/Si electrodes for the removal of macropollutants present in LLs under different electrochemical conditions, and (II) to recognize the dependencies between the reaction parameters affecting the COD and N-NH_4_^+^ removals (e.g., by-product formation). To gain a better understanding of the contaminants’ degradation kinetics in LLs samples by EO method, the different BDD/Si electrodes (levels of boron doping [B]/[C] = 500, 10,000, and 15,000 ppm—0.5 k; 10 k and 15 k, respectively) were studied. To date, the EO of old LLs has not been tested yet in laboratory-scale experiments with multiple iterations and a wide range of j values tested (in the range from 25 to 100 mA·cm^−2^). The EO reaction rates have been quantitatively discussed in order to optimize reaction condition. Furthermore, emphasis was placed on minimizing process costs and the preparatory protocol for carrying out the EO of LLs. Energy of removal of contaminants was calculated and optimized, additionally. Hence, this study can help to find the optimal conditions for the most cost-effective treatment of LLs with EO/BDD process.

## 2. Materials and Methods

### 2.1. LL Characteristics

The raw LLs used in this study were collected from the “Eko Dolina Lezyce” Municipal Solid Waste Plant (MSWP). The MSWP is located in the Pomerania region of north Poland, in Central Eastern Europe. The samples were collected from a closed, stabilized landfill cell that was operated from 2003 to 2011. While in operation, the landfill received mainly municipal waste, with unlimited disposal of organic wastes, including multi-material packaging or plastics [29].

The sanitary LLs used in this study were collected in 2019, before each experiment. Collected samples were transported in polyethylene containers (three bottles with a volume of 10 L each) to the laboratory (at 4 ± 1 °C). The samples were homogenized, then their physicochemical properties were evaluated. Each measurement was made three times.

The investigated raw LLs were rich in nitrogen (Total nitrogen, TN = 2148 ± 108 mg·L^−1^ and N-NH_4_^+^ = 2069 ± 103 mg·L^−1^) and organic substances (COD = 3608 ± 123 mg O_2_·L^−1^ with relatively low values of BOD_20_ = 403 ± 54 mg O_2_·L^−1^). The low BOD_20_/COD ratio (0.12 ± 0.00) obtained is regarded as typical for LLs generated by old landfill cells. Furthermore, LLs in this study were characterized by low TSS (<90 mg·L^−1^) and high N-NH_4_^+^/TN ratio (from 0.95 to 0.98). All obtained data indicated that the tested LLs were refractory to biological treatment. On the other hand, inter alia, high concentrations of chloride (2690 ± 70 mg Cl^−^·L^−1^), sulfate (1353 ± 70 mg SO_4_^2−^·L^−1^), and electrolytic conductivity ranging from 21.2 to 26.8 mS·cm^−1^ indicated the possible high efficiency of the EO process, if used for LL treatment.

### 2.2. Boron-Doped Diamond Electrodes (BDDs)

The BDD electrodes were synthesized in the Microwave Plasma Assisted Chemical Vapor Deposition (MWPACVD) process (SEKI Technotron AX5400S). Diamond films were deposited on two-inch Silicon wafers (ITME, Warsaw, Poland). The chamber pressure was kept at 50 Torr and the plasma was generated with microwave radiation (2.45 GHz) 1300 W. The induction heating stage temperature was set to 700 °C, the total flow rate of gases reached 300 sccm and the molar ratio of methane was equal to 1%. Three different levels of boron doping in the gas phase, expressed as [B]/[C] ratio, was 500, 10,000, and 15,000 ppm using diborane (B_2_H_6_) as a dopant precursor. In total, three different electrodes were prepared. The deposition time was set at 12 h.

### 2.3. Experimental Apparatus and Procedure

In regard to a low amount of total suspended solids (70.8 ± 10.0 mg·L^−1^) in the tested samples no pre-treatment nor pre-filtration was needed to conduce the electrolysis process. An LLs sample dilution of 1:1 (*V:V*) was applied. Electrooxidation runs were performed using 400 mL of sample in a 500-mL single-chambered reactor equipped with a magnetic stirrer (Electrochemical Stirrer, ES24, Wigo, Pruszkow, Poland) with a stirring speed of 300 rpm to keep the wastewater homogeny mixed. The temperature of the solution was maintained within 25 ± 1 °C (cooling bath). Assays were performed in the galvanostatic mode (power supply GW Instek GPD-23035, New Taipei City, Taiwan). Figure 1 shows the procedure used.

Four different j of between 25 and 100 mA·cm^−2^ and three different BDD electrodes as anodes named 0.5 k, 10 k, and 15 k were tested (see point 2.2). Stainless-steel mesh served as the cathode. Both the anode and the cathode were used as flat with a geometric surface area of 10.5 cm^2^ and the interelectrode distance ca. 2.0 cm. All assays were replicated three times, and samples (15 mL each) were collected every 2 h, until a final time of 8 h. Samples were degassed by mixing on a multipoint stirrer (Variomag, POLY 15 KOMED, Thermofisher Scientific, Port Orange, FL, USA) at 50 rpm for 15 min. This was necessary in order to perform a correct physico-chemical analysis of the samples. Table 1 presents a list of the physico-chemical analysis carried out in LLs, the methods of determinations used and the laboratory equipment used (all according to the APHA 2005 standard).

### 2.4. Calculation

#### 2.4.1. The Energy Consumption

One of the main challenges for AOPs is to reduce the energy consumption [36]. The energy consumption was expressed as W, kWh and was calculated in the following way: the applied current [A] was multiplied by electrolysis time [8 h] and the average cell voltage (E_cell_) [V], then this value (W, kWh) was recalculated and expressed in kWh·m^−3^. The specific energy consumptions in [kWh·kg^−1^ COD] and [kWh·kg^−1^ N-NH_4_^+^] was estimated by means of Equations (3) and (4):EC_COD_ = (1000 × E_cell_ × I × Δt)/(Vs [ΔCOD])(3)
EC_N-NH_4+__ = (1000 × E_cell_ × I × Δt)/(Vs [ΔN-NH4^+^])(4)
where: 1000 is a conversion factor (in mg·g^−1^), E_cell_ is the average cell voltage (in V), I is the applied current (in A), Δt is the electrolysis time (in h), Vs is the LL volume (in L), and (ΔCOD) is the experimental COD concentration decay (in mg·L^−1^).

Additionally instantaneous current efficiency (%ICE) was calculated (see Table A1) according to the following Equation (5):
%ICE = 100% × F × Vs × (COD_0_ − COD_t_)/ (m × I × Δt)(5)where: F is the Faraday constant (96,485 C mol^−1^), Vs is the total volume of the bulk electrolyte (L), COD_0_ and COD_t_ (in gO_2_L^−1^) are the initial and final COD obtained before and after EO treatment, I is the current (A), m is the equivalent mass of oxygen (m = 8), and ∆t is the electrolysis time (s).

#### 2.4.2. Kinetic Study

The order of the reaction of removing selected pollutants from the LLs was determined graphically using Statistica 12 (StatSoft, Inc., Tulsa, OK, USA). To determine the order of reactions with regard to reactant, three plots were made (concentration of contaminant versus time—zeroth-order reaction, natural log of concentration versus time—first-order reaction and inverse concentration versus time—second-order reaction). The most linear graph (linear regression was used) with the highest coefficient of determination value showed the order of the reaction with regard to reactant. After determining the orders of reactions, the rate constants and half-life were calculated using the appropriate kinetic equations.

#### 2.4.3. Biodegradability Index

Biodegradability indexes (BI) as BOD_20_ to COD ratios were determined for raw LL samples (before the electrochemical oxidation process) and for all samples after 8 h of treatment by means of EO (different current densities and different boron concentrations in anodes used).

### 2.5. Data Analysis

All plots and statistical analysis of data were performed using Statistica 12 (StatSoft, Inc., Tulsa, OK, USA), Microsoft Excel^®^ 2016 (Microsoft, 2016, Redmond, Washington, USA), and OriginPro 9.0 (OriginLab Corporation, Northampton, MA, USA) software.

## 3. Results and Discussion

### 3.1. Electrodes Morphology and Composition

The deposited electrode with various boron content strongly influences surface morphology. The average grain size for the 0.5 k electrode was approximately 2 µm. In the case of highly boron-doped 15 k, the grain size decreased to approximately 0.5 µm. Boron introduces re-nucleation which results in the creation of smaller crystallites on primary higher diamond crystals. This significant change in surface morphology impacts film composition results with more non-diamond content [37], which also increases internal stress in the film [39]. Moreover, increasing boron level in the BDD films influences their electro-catalytic performance, resulting in time decrease of the electrolysis [40,41] (stronger generations of ^^•^^OH radicals), but, in contrast, it reduces the width of the potential working window.

Experimental Raman spectra are shown in Figure 2. The sp^3^ diamond lines are located at 1332, 1317, and 1310 cm^−1^ for the 0.5 k, 10 k and 15 k, respectively. The boron-doping affects the diamond line causing a shift towards lower wavenumbers. Additionally, for the 10 k and 15 k, the sp^3^ peak is strongly asymmetric which is attributed to the Fano effect [42], a result of interference between the scattering by the zone-center phonon line and the scattering by an electronic continuum [43,44]. It is worth noting that high levels of doping reveal two broad peaks located at ca. 500 and 1200 cm^−1^, which are attributed to the incorporation of boron in the diamond lattice [45]. According to the literature, these two broad bands are attributed to the maxima of the phonon densities of the states [46]. Only for the lowest doped sample (0.5 k), peaks assigned to the first order of silicon are visible, located at 520 cm^−1^ for one-phonon mode, and the second-order of silicon is visible at 970 cm^−1^.

Additionally, the Fano-shaped peaks located at ca. 500 cm^−1^ (BWF function #1), 1200 cm^−1^ (BWF function #2), and 1332, 1317, and 1310 cm^−1^ (BWF function #3) were modelled using the Breit–Wigner–Fano function (see Equation (6)) [47] which is shown in Figure 3a–c insets.
(6)$$F_{i}\left(\omega \right)=\frac{A_{i}\times {\left(q_{i}+\frac{\omega -\omega_{i}}{\Gamma_{i}}\right)}^{2}}{1+{\left(\frac{\omega -\omega_{i}}{\Gamma_{i}}\right)}^{2}}$$

In the equation, *A_i_* is the amplitude of the Fano-shaped peaks, *q_i_* is the asymmetric parameter, *ω**_i_* is the width, and *Γ_i_* is the position of the lines.

The fitting parameters are shown in Table 2. In the case of the 0.5 k, the sample results in low boron incorporation into the diamond lattice that in turn results in lack of BWF#1 and BWF#2 bands. The amplitude of the Fano-shaped sp^3^ (BWF#3) peak is lower for the highly doped samples, but the peak width (ω) is wider, which is attributed to more boron incorporation into the diamond lattice [43]. According to an investigation by Mortet and co-authors [47], the asymmetric parameter of the diamond sp^3^ line (q_3_) can be used as a marker of the carrier concentration. Only the 10 k and 15 k samples indicated significant asymmetry (10 k q_3_ = 1.77 and 15 k q_3_ = 2.11), and it should be noted that the q_3_ value of those electrodes proves high incorporation of boron atoms during MWPACVD growth.

Additionally, the band located at ca. 500 cm^−1^ can be used to calculate the amount of boron in the diamond film. In the case of the 10 k electrode, the boron content is ca. 8.32 × 10^20^ and in the case of the 15 k, it is 1.05 × 10^21^. The slight difference in boron amount between 10 k and 15 k can be explained by a decrease in the effective doping of the diamond films during growth and an increase in the amount of boron in the gas phase.

The measurement of the electrochemical potential windows was carried on in three electrode setup in 1 M KCl at scan rate 100 mV/s, where the BDD electrode was as working, the Pt wire as counter and Ag/AgCl as reference. Figure 4 shows electrochemical potential windows of BDD electrodes with different boron doping. The increasing amount of boron content in diamond films narrows the width of electrochemical window. The 0.5 k doped sample have the wider window reaching up to 3.9 V, followed by 10 k sample with 3.12 V. The narrow window for the 15 k BDD, 2.23 V is connected with significant amount of defects in the diamond films.

### 3.2. COD Electrooxidation

In this study, during EO of LLs, in general the COD and N-NH_4_^+^ elimination rate increased with increasing j and time, which is well in agreement with the literature [48,49,50]. However, the complete removal of studied contaminants was not reached. It was noticed that, using j = 50–100 mA·cm^−2^, the increase in efficiency of COD removal was not as substantial as when using a lower j (see Figure 5a). It is generally considered that applying high j leads to the transfer of hydroxyl radicals (^^•^^OH) to H_2_O_2_ near the electrode surface, and consequently H_2_O_2_ is oxidized to O_2_ [51]. Moreover, the energy is consumed not mainly for oxidation of organic compounds (expressed as COD), but to a large degree for oxidation of other ions in the LLs [52].

The 0.5 k anode was the most effective electrode with C_8h_/C_0_ = 0.09 ± 0.14 for COD and 0.14 ± 0.01 for BOD_20_ after 8 h of process, at 100 mA·cm^−2^. In the same conditions, using 15 k, C_8h_/C_0_ was 0.27 ± 0.31 for COD, and 0.23 ± 0.08 for BOD_20_, while for 10 k, C_8h_/C_0_ was 0.3 ± 0.001 for COD and 0.26 ± 0.01 for BOD_20_. COD was most effectively removed during the first two hours of the EO process, although removal efficiency decreased with time (for all tested electrodes and j).

EO treatment can also improve the biodegradability of LLs by oxidizing the molecular structure of refractory organics, and thus can be used as a pre-treatment process before, for example, further biological treatment. However, it should be noted that the best conditions ensuring good COD removal are BI > 0.5 [53]. Fernandez and co-authors [21,53], when testing the EO (BDD anodes, j = 700 mA·cm^−2^, 36 h of process) for the treatment of sanitary LLs obtained an increase in biodegradability index (BI) from 0.18 to 0.84. Meanwhile, Nurhayati [54] got an increase in BI ranging from 0.06 in raw LLs up to 0.386 after electrooxidation process at j of 30 mA·cm^−2^ with a flow rate of 5 mL·s^−1^. In this study, the calculated BI of raw and EO treated LLs is presented in Figure 6. BI in tested raw samples was equal to 0.11 ± 0.13.

Interestingly, the use of the highest j (100 mA·cm^−2^) resulted in obtaining a BI in the treated LLs close to that calculated for the raw sample (BI for 0.5 k = 0.11 ± 0.00, 10 k = 0.097 ± 0.005, and 15 k = 0.090 ± 0.01). This can be explained by the large share of non-biodegradable intermediates in the sample. The highest BI = 0.22 ± 0.05 was obtained using a 0.5 k electrode and a j of 50 mA·cm^−2^, successively using a 10 k electrode BI = 0.16 ± 0.01 and a current of 30 mA·cm^−2^. Hence, following Deng and co-authors as well as McBeath and co-authors [20,55] in applying such a condition, EO might be used prior to biological treatment due to the doubling of BI. Generally, the results showed that BDD anodes are very effective in oxidation of organic matter, what can be attributed, inter alia, to their high oxygen evolution potential.

### 3.3. N-NH_4_^+^ Electrooxidation

For all anodes tested in this study, N-NH_4_^+^ removal was less than COD removal (e.g., 0.5 k, j = 100 mA·cm^−2^, C_8h_/C_0_: COD = 0.09 ± 0.14 mg·L^−1^, and N-NH_4_^+^ = 0.39 ± 0.05 mg·L^−1^). Other researchers obtained similar results. After 6 h of process, Zhou and co-authors [56] achieved 87.5% COD and 74.06% N-NH_4_^+^ removal by the BDD/Nb electrodes using j = 50 mA·cm^−2^. Luu and co-authors [57] explored EO of biologically treated LLs using Ti/BDD and Ti/RuO_2_ anodes and observed the maximum COD and N-NH_4_+ removal efficiency of 95.17% and 81.18%, respectively (after 8 h, at 83 mA·cm^−2^). This phenomenon is characteristic of EO of LLs containing Cl^−^ in the range of concentrations from 150 to 4500 mg Cl^−^·L^−1^ by means of BDD anodes (primarily COD is oxidized) [58]. Ammonia removal is mainly promoted by a reaction with active chlorine (chlorine-mediated pathway: Cl_2_/HOCl^—^ indirect oxidation) and the oxidization rate of N-NH_4_^+^ by hydroxyl radicals is lower than that of organics (“electrochemical combustion” process) [20]. Furthermore, BDD anodes are more suitable for ^^•^^OH radical generation than chlorine evolution.

Regarding N-NH_4_ and TN removals from LLs, it can be assumed that j is a key experimental parameter affecting this process (Figure 5b and Figure A3 (Appendix A)). The increase in j (according to Faraday’s law) enhanced the electrochemical process and resulted in a satisfactory result in N-NH_4_^+^ and TN removal [59,60].

Removal of N-NH_4_^+^ is also strongly pH-dependent [20]. It was found that ammonia ions present in the initial sample were partially transformed to the nitrate or nitrite forms. However, based on Figure 7, it might be assumed that some ammonia nitrogen was transferred directly to gaseous nitrogen—approximately 417, 385, and 294 mg N-N_2_·L^−1^ for BDD 0.5 k, 10 k, and 15 k, respectively.

Application of high current densities led to a fast increase in pH to above 8, which promoted the direct oxidation of free ammonia (gaseous) according to reactions (7) and (8) [61]:
2NH_3_ + 3H_2_O→NO_3_^−^ + 9H^+^ + 8e^−^(7)

2NH_3_→N_2_ + 6H^+^ + 6e^−^(8)

According to Zhou and co-authors [56], it might be assumed that higher ammonia removal efficiency might be achieved in acidic pH (where HOCl^−^ are predominant species). Furthermore, removing ammonia from LLs via an active chlorine-mediated (indirect) path can be effective only if it occurs quickly, so good results can be obtained inter alia in high temperatures [62]. If these conditions are not met, the active chlorine is often converted to chlorate and perchlorate, and ammonium nitrogen is not removed effectively.

### 3.4. The Evolution of Nitrates and Nitrites during EO

Figure A1 and Figure A2 (Appendix A) report the change of nitrate and nitrite concentration with time. In general, the concentration of N-NO_3_^−^ increased intensively (or remained at the same level) during the whole electrochemical process due to partial oxidation of organic matter containing nitrogen and ammonia. For instance, at a j of 100 mA·cm^−2^ C_8h_/C_0_ for N-NO_3_^−^ was 42.6 ± 3.0, 6.0 ± 1.8, and 39.2 ± 9 for 0.5 k, 10 k, and 15 k, respectively.

A similar dependency occurred in the case of N-NO_2_^−^ concentrations (from 0.20 mg·L^−1^ up to 8.95 mg·L^−1^ for 0.5 k, j = 100 mA·cm^−2^). N-NO_2_^−^ ions were possibly formed through the EO of ammonia on the anode (Equation (9)) or electrochemical reduction of nitrate on the cathode (Equation (10)) [63]. However, oxidation of amine groups present in organic compounds needs to be taken into consideration during NO_2_^−^ generation. The amine group in organic functional group redox series is very reductive and easily oxidized [64]:
NH_4_^+^ + 2H_2_O→ NO_2_^−^ + 8H^+^ + 6e^−^(9)
NO_3_^−^ + H_2_O + 2e^−^→NO_2_^−^ +2OH^−^(10)

It should also be noted that, in our study, TN was mainly composed of N-NH_4_^+^. For this reason it was concluded that the oxidation of organic nitrogen had a negligible effect on the effects of TN removal in the tested samples.

### 3.5. Changes and Influence of pH on Electrooxidation

The initial pH of the tested raw LLs was 7.8 ± 0.1 and generally, with time of EO treatment, the pH of the LLs also increased (Figure A4, Appendix A). The high concentration of carbonate/bicarbonate ions in raw LLs, which are effective ^^•^^OH radical scavengers, may cause an increase in pH during EO treatment [65]. At the same time, the carbonate–bicarbonate equilibrium (Equation (11)) is shifting to the right.


CO_2_ + H_2_O ↔ H^+^ + HCO_3_^−^ ↔ 2H^+^ + CO_3_^2−^(11)


In this study, the exceptions were observed during EO processes using 0.5 k and 10 k electrodes with j = 25 mA·cm^−2^ and 10 k, 15 k using j = 30 mA·cm^−2^. In these cases, a decrease in pH was observed with time. This phenomenon is supposed to be related to the electrochemical reactions intensively taking place on the surface of the anode [34,66] (please see Equations (12) and (13)). Another option could be that low-molecular-weight carboxylic acids present in LLs [67,68] undergo the following reactions presented in Equations (14) and (15) [69].


H_2_O → ^•^OH + H^+^ + 1e^−^(12)
2H_2_O → O_2_ + 4H^+^ + 4e^−^(13)
HCOOH → 2CO_2_+ 2H^+^ + 2e^−^(14)
CH3COOH carboxylic acids + 2H_2_O → 2CO_2_+ 8H^+^ + 8e^−^(15)


On the basis of the obtained pH results, it was also found that the HOCl^−^ ion could play a large part in the oxidation of N-NH_4_^+^ (especially during the first 4 h of the oxidation process). Higher j shifted the pH to higher values, which led to the formation of the hypochlorite anion (ClO^−^), the presence of which induced the formation of toxic chloramines and poor oxidation properties. It is supposed that one of the main reasons for the increase in pH was the reactions forming, inter alia, hydroxyl ions during the electrochemical reduction of nitrate [70]. Another reason could be the intense generation of sulfate radicals (reaction of the sulfate present in the LLs with the hydroxyl radicals). This reaction contributes to increasing the concentration of hydroxyl ions in LLs and increasing the pH value (Equations (16) and (17)) [71]:
SO_4_^2−^ + ^•^OH → SO_4_^•−^ + OH^−^(16)
2H_2_O + 4e^−^→ H_2_ + 2OH^−^(17)

However, it should be noted that at pH = 8 sulfate and chloride radicals are not permanent and the equilibrium is shifted in the ^^•^^OH direction (Equation (15)). Moreover, the presence of chloride at the level of 2690 ± 70 mg·L^−^^1^ might have an ‘inhibiting effect’ on organics removal by sulfate radicals, because chlorine reactive species are expected to be the dominant oxidative species in the anodic oxidation of LLs [52].

### 3.6. COD and N-NH_4_^+^ Kinetic Evaluation

The results of the graphical method of the reaction kinetic order evaluation are presented in Figure 8 and Table 3. It should be noted that the rate constants in all cases (k) showed great dependence on the applied j: generally it increased with increasing applied j, and the value of the half-life (T_1/2_) consistently decreased (Table 3). Ukundimana and co-authors [25] reported that a mechanism of COD removal from LLs (pre-treated in an ultrafiltration unit) by means of BDD electrodes fitted well with the pseudo-first-order kinetic model. A similar phenomenon has been observed in another study, in which Papastavrou and co-authors [72] treated stabilized LLs by means of BDD anodes and obtained the pseudo-first-order kinetic model for COD removal with a kinetic coefficient of 8.3 ± 1·10^−3^·min^−1^ for current values 15 and 21 A. Conversely, studies by Cossu and co-authors [73] demonstrate that the rate constants (EO of LLs; PbO_2_, and SnO_2_ anodes) decrease over time due to the presence of compounds that are more easily oxidized than others in the initial LL sample. Another explanation could be that due to the very complex matrix of LLs (containing e.g., high-molecular-weight compounds such as humic and fulvic acids, and recalcitrant substances with a low molecular weight such as halogenated compounds), some compounds are more easily oxidized than others during the first stage of electrolysis [74].

According to the literature, kinetics of N-NH_4_^+^ removal from LLs usually shows sigmoidal ammonium concentration profiles and the order of reaction perfectly describes Equation (18) [75]:(18)$$\frac{d{\left[N-NH_{4}\right]}^{+}}{dt}=\;-k\;{\left[N-NH_{4}\right]}^{+}\times \left[{\mathrm{Cl}}_{2}\right]$$
where: [*N-NH_4_*]^+^ is the ammonia concentration, *k* is the second-order rate constant, and [Cl_2_] is the concentration of dissolved active chlorine.

The results of this study showed that after 8 h of treatment with applied j of 100 mA·cm^−2^, the average C_8h_/C_0_ for TN was 0.60, 0.66, and 0.75 using 0.5 k BDD, 10 k BDD, and 15 k BDD, respectively. In the same conditions, C_8h_/C_0_ for N-NH_4_ was 0.39 (0.5 k BDD), 0.63 (10 k BDD), and 0.58 (15 k BDD). Figure 8 and Table 3 indicate that, in most cases, ammonia removal in different j followed second-order kinetics, but also first- and zeroth-order.

The differences in the trend of ammonium removal depended on the applied j and the electrode used, and might be attributed, inter alia, to the existence of different forms of active chlorine in different pH values. These results are in agreement with the results obtained by other authors [30,76,77,78]. Cabeza and co-authors [78,79] studied the EO of LLs by means of BDD electrodes using j ranging from 150 to 900 mA·cm^−2^. The results showed that high values of the applied j caused elimination of N-NH_4_^+^ with zeroth-order kinetics, whereas, using low values of the applied j, exponential-like decaying ammonia concentration curves were obtained. On the other hand, according to Li and co-authors [50], removal of N-NH_4_^+^ from LLs by means of Ti/RuO_2_–IrO_2_ and Al electrodes indicated that the second-order equation fitted well. Basically, as in the case of COD removal, the rate constant of N-NH_4_
^+^ increased with increasing applied j, and the value of the half-life consistently decreased. The nitrogen was best removed by a 0.5 k electrode, which, due to having the lowest boron-doping level, was the most efficient in direct ammonium oxidation. The similar phenomenon was observed during PFOA and PFOS oxidation, where direct electrochemical reaction related with the perfluoroalkoxy radicals and hydroxyl radicals “production” on the electrode occurs more intensively on 0.5 k Nb/BDD than on 10 k Nb/BDD [29].

### 3.7. Energy Consumtion Optimisation

High energy consumption (EC) is one of the major drawbacks of EO, so energy consumption analysis should be an inherent part of all research scientific publications on the EO issue. Developing cost effective and stable anodes and application of renewable energy sources in EO treatment of LL is currently a major challenge [80]. Figure 9 shows the EC after 8 h of assays (with varying j application). In this study, EC_COD_ and EC_N-NH4+_ increased and ICE decreased with increasing j and k (see also Figure 10 and Table A1), which is in good agreement with the literature [81,82,83,84].

For example, Panizza and co-workers [85] indicated an increase in energy consumption with increased j and achieved complete removal of COD after 7 h (flow rate 420 L·h^−1^, pH = 8.2, j = 100 mA·cm^−2^, PbO_2_) by anode with EC = 220 kWh·m^−3^. However, the initial concentration of the COD was low (780 mg·L^−1^) and the chlorides were present in a very-high-concentration Cl^−^ (1800 mg·L^−1^). In turn, the result of this study showed that the 15 k electrode consumed the lowest energy (expressed as kWh·kg^−1^ N-NH_4_^+^/COD), while the 0.5 k electrode consumed more, and the highest energy consumption was observed for the 10 k electrode (Figure 9 and Figure 10). The electrode that was most effective in removing contamination (BDD 0.5 k, j = 100 mA·cm^−2^, t = 8 h) had the following energy consumption: EC_COD_ = 285 kWh·kg^−1^ COD EC_N-NH4+_ = 748 kWh·kg^−1^ N-NH_4_^+^ and 495 kWh·kg^−1^ (Figure 9). These results are analogous to results obtained by our research group testing BDD/Nb electrodes in terms of LL treatment [29]. Figure 10 clearly shows that the energy consumption increased with increasing k, indicating that it appears to be more cost-effective to remove COD than N-NH_4_^+^ (using 0.5 k). However, thanks to such a wide database, optimal conditions might be proposed: the lowest energy consumption (94.5 kWh·m^−3^) for COD removal efficacy (>70%) may be achieved after 6 h of process using a 0.5 k electrode. Such a COD removal efficacy will be sufficient, e.g., in terms of EO application before biological treatment. Considering other types of electrodes (Ti/BDD and Ti/RuO_2_) operated at 83 mA/cm^2^ from 4 to 8 h, energy consumption ranged from 30 to 190 kWh·kg^−1^ COD. However, Tran Le Luu [57] pointed out that the energy consumption at Ti/BDD is less than Ti/RuO_2_ anode under such a condition concluded Ti/BDD anode as a more favorable than Ti/RuO_2_. Moreover, Zhou and co-authors [56] used ICE and EC to optimize the BDD/Nb EO process parameters. They showed that a current density of 50 mA/cm^2^ lead to 87.5% COD and 74.06% N-NH_3_ removal after 6 h of EO in flowing reactor with energy consumption of 223.2 kWh·m^−3^ and ICE equal to 35%. Under the same current density studied 0.5 k BDD/Si anode (after 8 h of process) in the batch reactors led to 83% COD and 48% N-NH_4_^+^ removal from LLs with energy consumption of 200 kWh·m^−3^, 122 kWh kg^−1^ COD, and ICE = 45% (Appendix A, Table A1).

Despite parameter optimization, other solutions that will decrease energy consumption of advanced oxidation processes have been greatly explored. For instance, Cardoso and co-authors [86] indicated that a bubbling reactor with an ozone system resulted in reduced energy consumption compared to a combination of ozone with photocatalysis or photo-electrocatalysis. The bubbling reactor might also be considered in the optimization of EO reactor efficacy, possibly resulting in less energy consumption.

## 4. Conclusions

The application of new approaches in advanced treatment technologies such as EO can deliver environmental, economic, and social benefits. To be in accordance with the EU Green Deal and Water-Smart Society concept, EO application for LL treatment should be optimized in a ‘fit-for-purpose’ manner that is also affordable and practical for water-related sectors. This study presents that EO using BDD anodes is an effective technology to treat LLs in terms of macropollutant removal. At an optimized test, applied j of 100 mA·cm^−2^ with the 0.5 k electrode, the electro-oxidation process could nearly completely remove COD from LLs, with C_8h_/C_0_ = 0.09 ± 0.14, but the removal of N-NH_4_^+^ was less effective (C_8h_/C_0_ = 0.39). Thus, the proposed treatment process is instead dedicated to COD degradation. In most cases, COD removal fits the model of pseudo-first-order reactions with good linearity. Some conditions lead to a two-fold increase in biodegradability index (BI = 0.22 ± 0.05 was obtained using a 0.5 k electrode and a j of 50 mA·cm^−2^). Such a phenomenon would suggest applying EO before biological treatment of LLs. EO of LLs may result in: (1) better subsequent effectiveness of biological processes due to the mineralization of poorly decomposable (non-biodegradable) COD and (2) an increase in N-NH_4_^+^ removal effectiveness.

The main drawback concerning applying the BDD electrodes for LLs treatment seems still to be the energy consumption. Considering using EO by means of BDD/Si in wider industrial and environmental applications appears complicated to implement for economic reasons. However, on the other hand, in this study, in a relatively short time, effective organic matter removal was obtained. Furthermore, the use of this method is also supported by the following factors: compact and modular reactor design, no addition of reagents needed, simple operation of devices, and the ability to adjust to variable organic loads in wastewater. To conclude, EO by means of BDD is an effective technology for the treatment of industrial wastewater such as LLs. Further studies are recommended to improve the ability of electrochemical conversion of ammonia to N_2_ in LLs (inter alia by controlling the main operating parameters). Thus, it is stated that new developments in nanotechnology and material science, the widespread use of alternative energy sources, and reductions in material costs for the production of the electrodes now appear to be crucial in the wider implementation of EO.

## Figures and Tables

**Figure 1 materials-14-04971-f001:**
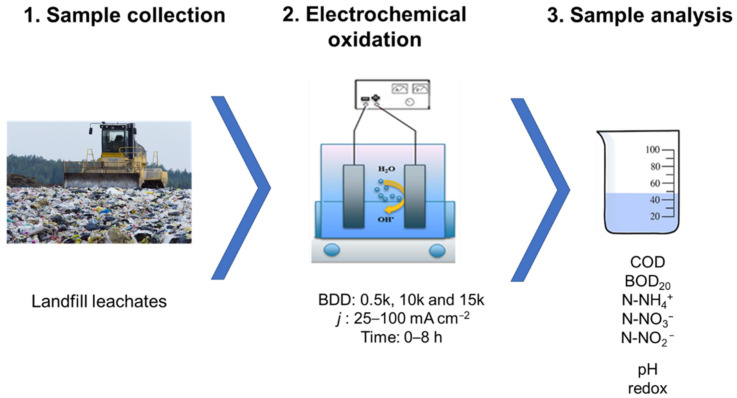
Procedure scheme.

**Figure 2 materials-14-04971-f002:**
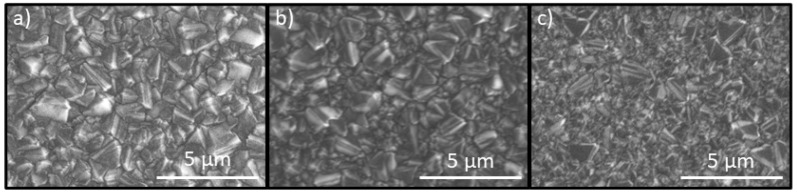
SEM images of BDD films deposited at different [B]/[C] ratios in plasma, namely, (**a**) 500 ppm, (**b**) 10,000 ppm, and (**c**) 15,000 ppm. Magnification of 10,000×.

**Figure 3 materials-14-04971-f003:**
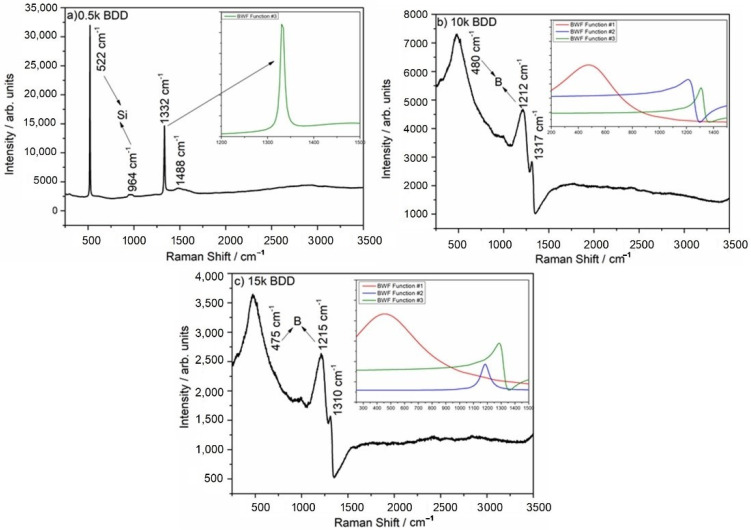
Raman spectra of (**a**) 0.5 k, (**b**) 10 k, and (**c**) 15 k BDD films. Insets shows Raman spectra modeled using Breit–Wigner–Fano function: Fano-shaped peaks (BWF #1 red line, BWF #2 blue line, and BWF #3 olive line).

**Figure 4 materials-14-04971-f004:**
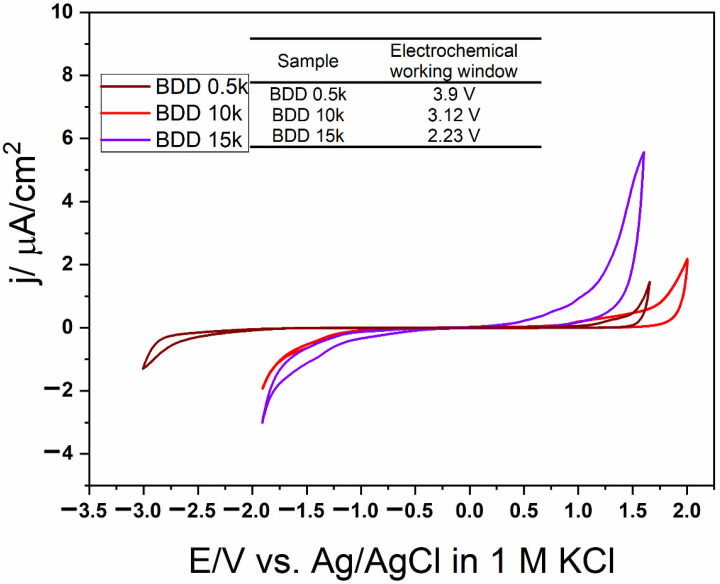
Cyclic voltammetry plots for BDD films with different boron doping recorded in 1 M KCl with 100 mV/s scan rate.

**Figure 5 materials-14-04971-f005:**
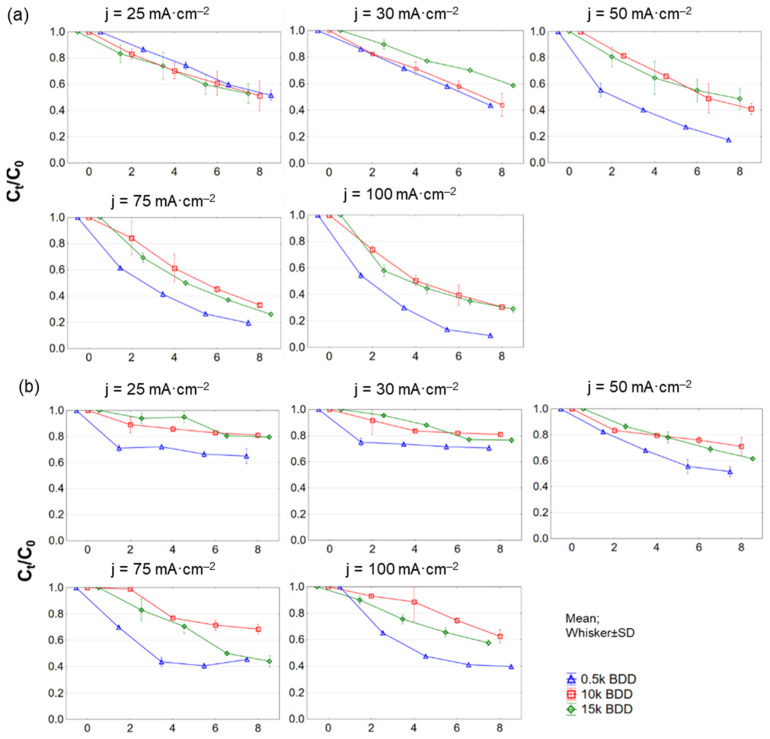
Removal of (**a**) COD and (**b**) N−NH_4_^+^ during 8 h test of LL treatment with different BDD anodes and current densities (25–100 mA·cm^−2^) expressed as normalized concentration C_t_/C_0_.

**Figure 6 materials-14-04971-f006:**
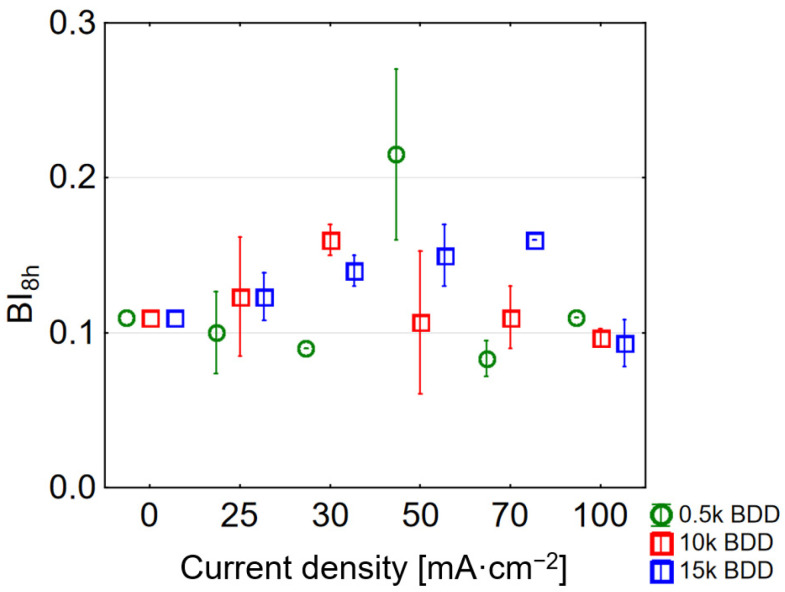
Influence of various current densities and electrodes used on BI (BI values after 8 h of LL treatment for different current densities applied).

**Figure 7 materials-14-04971-f007:**
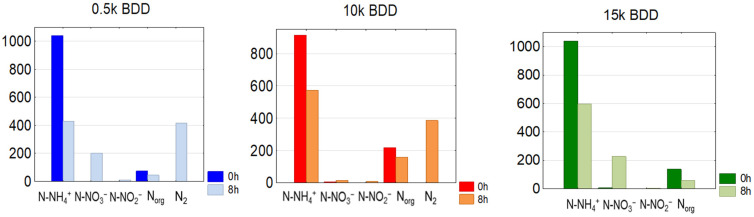
Average C_0_ and C_8h_ concentrations of nitrogen compounds in treated LLs by means of different anodes used, j = 100 mA·cm^−2^ (N_org_ = TN − (N-NH_4_^+^+ N-NO_3_^−^+ NO_2_^−^)).

**Figure 8 materials-14-04971-f008:**
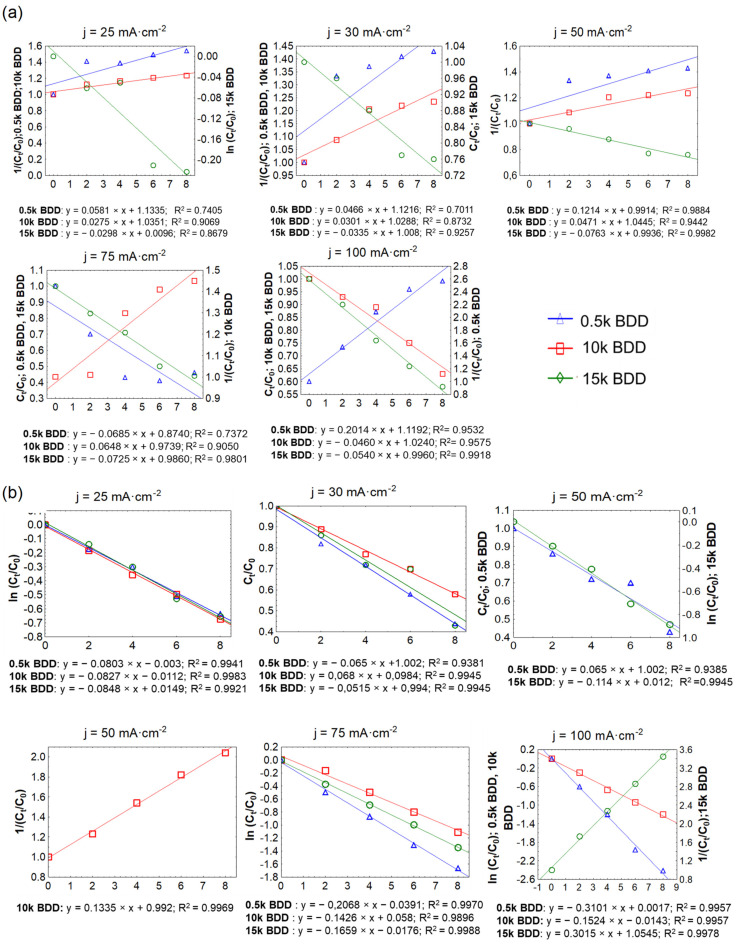
Determination of the order of removal of: (**a**) COD and (**b**) N-NH_4_^+^ by the graphical method.

**Figure 9 materials-14-04971-f009:**
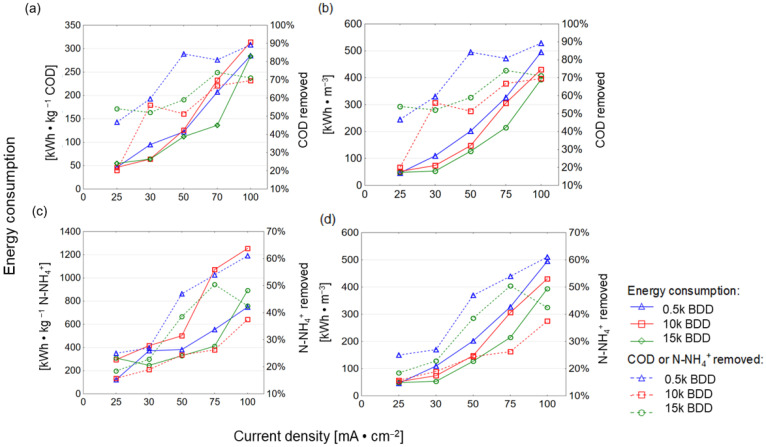
Energy consumption after 8 h of EO process using different BDD/Si electrodes and different current densities plotted against COD removed (**a**,**b**) and N-NH_4_^+^ removed (**c**,**d**).

**Figure 10 materials-14-04971-f010:**
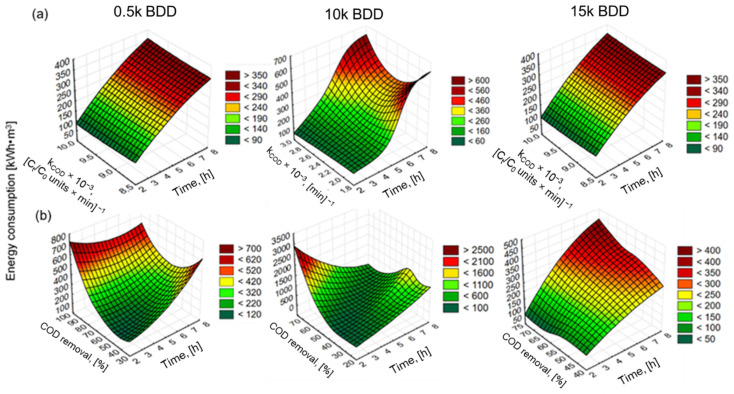
Surface plots (distance weighted least squares) of: (**a**) energy consumption against constant rate coefficient and time of EO and (**b**) energy consumption against COD removal efficacy and time of EO for j = 100 mA·cm^−^^2^ and different electrodes used.

**Table 1 materials-14-04971-t001:** List of the physico-chemical analysis applied.

Parameter	Method and Device Used
**COD, N-NH_4_^+^, N-NO_3_^−^,** **N-NO_2_^−^, TP, P-PO_4_^−^, Cl^−^, SO_4_^2−^, S^2−^**	XION 500 spectrophotometer (Dr Lange, GmbH, Düsseldorf, Germany)
**BOD_20_**	manometric respirometric BOD OxiTop^®^ method
**pH, conductivity and ORP**	portable multi-parameter meter (HL-HQ40d multi, HACH, Düsseldorf, Germany)
**total (TSS), mineral (MSS), volatile suspended solids (VSSs)**	the gravimetric method

**Table 2 materials-14-04971-t002:** Fitting parameters of BDD electrodes. Indices 1, 2, and 3 correspond to the BWF#1, BWF#2, and BWF#3 bands, respectively.

Parameter	0.5 k BDD	10 k BDD	15 k BDD
*A_1_*	-	3895	2232.13
*q_1_*	-	5.3	4.3
*ω_1_*	-	480.2	476.34
*Γ_1_*	-	248.70	236.81
*A_2_*	-	1309.63	1429
*q_2_*	-	0.88	0.46
*ω_2_*	-	1220	1214
*Γ_2_*	-	42.99	42
*A_3_*	11,860	1833	1039
*q_3_*	0.127	1.77	2.11
*ω_3_*	1332	1317	1309
*Γ_3_*	4.60	26.51	33.8

**Table 3 materials-14-04971-t003:** Electrochemical degradation kinetics for COD and N-NH_4_^+^ in LLs with different j and different electrodes used.

Parameter	Electrode	Current Density (mA·cm^−2^)	Order of Reaction	k *	T_½_ (min)
COD	0.5 k BDD	25	pseudo-first-order	1.34 ± 0.24 ×10^−3^	536.7 ± 124.7
		30	pseudo-zeroth-order	1.21 ± 0.01 ×10^−3^	412.2 ± 3.9
		50	pseudo-zeroth-order	1.72 ± 0.01 ×10^−3^	290.9 ± 2.4
		75	pseudo-first-order	3.46 ± 0.23 ×10^−3^	203.6 ± 13.5
		100	pseudo-first-order	4.76 ± 0.71 ×10^−3^	147.4 ± 2.2
	10 k BDD	25	pseudo-first-order	1.57 ± 0.36 ×10^−3^	459.6 ± 101.6
		30	pseudo-zeroth-order	1.19 ± 0.17 ×10^−3^	426.3 ± 61.9
		50	pseudo-second-order	2.38 ± 0.75 ×10^−3^	449.4 ± 146.2
		75	pseudo-first-order	2.30 ± 0.07 ×10^−3^	302.9 ± 9.5
		100	pseudo-first-order	2.47 ± 0.04 ×10^−3^	280.1 ± 5.5
	15 k BDD	25	pseudo-first-order	1.38 ± 0.16 ×10−3	506.7 ± 57.3
		30	pseudo-zeroth-order	8.80 ± 0.10 ×10^−4^	567.6 ± 4.5
		50	pseudo-first-order	1.86 ± 0.21 ×10^−3^	374.5 ± 43.4
		75	pseudo-first-order	2.80 ± 0.14 ×10^−3^	247.1 ± 12.1
		100	pseudo-second-order	5.22 ± 0.14 ×10^−3^	190.7 ± 12.3
N-NH_4_^+^	0.5 k BDD	25	second-order	1.09 ± 0.22 ×10^−3^	936.8 ± 191.3
		30	second-order	0.83 ± 0.07 ×10^−3^	1205.4 ± 98.7
		50	second-order	2.12 ± 0.09 ×10^−3^	472.1 ± 22.3
		75	zeroth-order	1.21 ± 0.01 ×10^−3^	413.6 ± 3.2
		100	second-order	3.41 ± 0.001 ×10^−3^	292.7 ± 8.5
	10 k BDD	25	second-order	0.49 ± 0.04 ×10^−3^	1958.2 ± 62.4
		30	second-order	0.51 ± 0.01×10^−3^	1959.0 ± 61.3
		50	second-order	0.84 ± 0.30 ×10^−3^	1267.8 ± 151.7
		75	second-order	1.09 ± 0.19 × 10^−3^	932.6 ± 161.4
		100	zeroth-order	0.80 ± 0.12 × 10^−3^	628.68 ± 96.6
	15 k BDD	25	first-order	0.48 ± 0.14 × 10^−3^	1450.9 ± 56.2
		30	first-order	0.55 ± 0.01× 10^−3^	1247.5 ± 129.0
		50	second-order	1.33 ± 0.01 × 10^−3^	749.6 ± 4.0
		75	zeroth-order	1.25 ± 0.01 × 10^−3^	399.3 ± 3.5
		100	zeroth-order	0.93 ± 0.05× 10^−3^	534.72 ± 26.7

k * units: 0th-order reactions = specific (C_t_/C_0_ units) × min^−1^, 1st-order reactions = min^−1^, 2nd-order reactions specific (C_t_/C_0_ units × min) ^−1^.

## Data Availability

Data sharing is not applicable to this article.

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
