# Peer review of "Kinetics of the Organic Compounds and Ammonium Nitrogen Electrochemical Oxidation in Landfill Leachates at Boron-Doped Diamond Anodes"

_materials, 2021, doi:10.3390/ma14174971_

Round 1

Reviewer 1 Report

Recommendation: Minor revision

The authors presented a study regarding the kinetics of the organic compounds and ammonium nitrogen 2 electrochemical oxidation in landfill leachates at boron-doped 3 diamond anodes (BDD/Si). This study is very comprehensive and systematic. The results are well-analyzed and scholarly presented. The conclusions are well-supported by the data. I suggest to publish this paper in Materials after the following minor issues are clearly addressed:

  1. The significance and potential medical values of using electro-oxidation to treat the organic compounds and ammonium in the matrix should be more emphasized in the introduction;
  2. Figure 2 should be re-drawn. It is not clear, and the y axis is elongated, high-resolution image must be provided;
  3. For Table 2, the current density of 25 mA cm-2 is too close to 30 mA cm-2, this interval is too close, so 15 mA cm-2 or 20 mA cm-2 should be considered;
  4. Figure A1 is too crowded to observe the figures, it can be divided into 3 figures to ensure it can be seen clearly.

In summary, this work is of some scientific merits, and after the above issues are unambiguously addressed, it is suggested to be published in Materials.

Reviewer 2 Report

Wilk et al investigated the "Kinetics of the organic compounds and ammonium nitrogen electrochemical oxidation in landfill leachates at boron-doped diamond anodes". Various physical and electrochemical characterizations of BDD were performed. And the obtained results, At an optimized test, applied j of 100 mA·cm˗2 with the 0.5k electrode, the electro-oxidation process could nearly completely remove COD from LLs. I recommend the publication of the work after the minor comments as stated below. 

  1. Authors should compare the energy consumption of the present work with the existing literature? for example, Ref. 58 and 59. Did the authors able to reduce the energy consumption? If not, what are the possible strategies that could be suggested? 
  2. BDD possesses a wide range of potential windows. I would love to see the potential window of the different B doping sample electrodes performed in this study (by means of a cyclic voltammogram).
  3. A graphical representation of energy consumption (ICE) vs energy consumption should be presented for different B doping sample electrodes performed in this study
  4. What is the effect of higher or lower rpm on the removal rate? is there any practical effect ? or was it a randomly chosen parameter?
  5. All the figure's quality should be improved. It's hard to read the X and Y axis values. The line thickness, font size, etc should be increased for clear visibility.
